# Comprehensive characterization of mitochondrial bioenergetics at different larval stages reveals novel insights about the developmental metabolism of *Caenorhabditis elegans*

**Danielle F. Mello**[¤][⊙], **Luiza Perez**[⊙], **Christina M. Bergemann**[⊙], **Katherine S. Morton**[⊙], **Ian T. Ryde**, **Joel N. Meyer** *

Nicholas School of the Environment, Duke University, Durham, North Carolina, United States of America

⊙ These authors contributed equally to this work.
¤ Current address: Ifremer, CNRS, IRD, LEMAR, IUEM, Univ Brest, Plouzane, France
* joel.meyer@duke.edu

## Abstract

Mitochondrial bioenergetic processes are fundamental to development, stress responses, and health. *Caenorhabditis elegans* is widely used to study developmental biology, mitochondrial disease, and mitochondrial toxicity. Oxidative phosphorylation generally increases during development in many species, and genetic and environmental factors may alter this normal trajectory. Altered mitochondrial function during development can lead to both drastic, short-term responses including arrested development and death, and subtle consequences that may persist throughout life and into subsequent generations. Understanding normal and altered developmental mitochondrial biology in *C. elegans* is currently constrained by incomplete and conflicting reports on how mitochondrial bioenergetic parameters change during development in this species. We used a Seahorse XFe24 Extracellular Flux (XF) Analyzer to carry out a comprehensive analysis of mitochondrial and non-mitochondrial oxygen consumption rates (OCR) throughout larval development in *C. elegans*. We optimized and describe conditions for analysis of basal OCR, basal mitochondrial OCR, ATP-linked OCR, spare and maximal respiratory capacity, proton leak, and non-mitochondrial OCR. A key consideration is normalization, and we present and discuss results as normalized per individual worm, protein content, worm volume, mitochondrial DNA (mtDNA) count, nuclear DNA (ncDNA) count, and mtDNA:ncDNA ratio. Which normalization process is best depends on the question being asked, and differences in normalization explain some of the discrepancies in previously reported developmental changes in OCR in *C. elegans*. Broadly, when normalized to worm number, our results agree with previous reports in showing dramatic increases in OCR throughout development. However, when normalized to total protein, worm volume, or ncDNA or mtDNA count, after a significant 2-3-fold increase from L1 to L2 stages, we found small or no changes in most OCR parameters from the L2 to the L4 stage, other than a marginal increase at L3 in spare and maximal respiratory capacity.

**Data Availability Statement:** All relevant data are within the manuscript and its Supporting information files.

**Funding:** This work was funded by the National Institute of Health (R01ES028218, P42ES010356, R01ES034270, 2T32ES021432). The grants were awarded to JNM. Some strains were provided by the Caenorhabditis Genetics Center, which is funded by NIH Office of Research Infrastructure Programs (P40 OD010440). The funders had no role in study design, data collection and analysis, decision to publish, or preparation of the manuscript.

**Competing interests:** The authors have declared that no competing interests exist.

Overall, our results indicate an earlier cellular shift to oxidative metabolism than suggested in most previous literature.

## 1. Introduction

Mitochondria play key roles in development and disease [1–3]. Understanding how genetic and environmental factors impact mitochondrial function is critical, because ~1 person in 4,000 suffers from a mitochondrial disease [4], and many pollutants [5] and drugs [6] cause mitochondrial toxicity. Mitochondria are also key mediators of cellular stress and immune responses [7].

The model organism *Caenorhabditis elegans* has been used extensively to study developmental biology and genetics [8], mitochondrial disease [9], and mitochondrial toxicity [10]. *C. elegans* development includes the egg stage (in which ~50% of the final number of somatic cells are made), followed by 4 larval stages and adulthood. Early work by Lemire and colleagues demonstrated that pharmacologically inhibiting mitochondrial function during development blocked or slowed developmental progression from the L3 to the L4 larval stages [11], as did blocking mitochondrial DNA (mtDNA) replication pharmacologically [12]. Strains carrying mutations in genes encoding mitochondrial proteins [11] or the DNA polymerase responsible for mtDNA replication of [13] also showed slowed or blocked development, as did worms exposed to a variety of pollutants that targeted mitochondria [14, 15]. Lower levels of mitochondrial stress resulting from RNA inhibition that still permit development to adulthood can lead to both deleterious and beneficial outcomes later in life [16–18]. Similarly, both deleterious and beneficial outcomes later in life can result from lower levels of mitochondrial stress induced by chemical exposures during development [19, 20]. Mitochondrial toxicity also affects *C. elegans* neuronal function [21, 22], immune function [23, 24], and other cellular processes [25–27] impacted by mitochondrial dysfunction in higher eukaryotes. However, while *C. elegans* is generally an excellent model for studying how genetic and environmental factors affect mitochondrial function during development, this utility is limited by incomplete and in some cases conflicting literature reports of how mitochondrial biology changes during development in this organism.

The integrity and transcription of the mitochondrial genome is necessary for aerobic energy production, because mtDNA encodes 12 (in worms) or 13 (in humans) core proteins of the mitochondrial respiratory chain (MRC). The mtDNA copy number (mtDNA CN, or number of mitochondrial genomes per cell) is highly variable by cell type, and undergoes a developmental bottleneck because mtDNA is not replicated in early stages after oocyte fertilization, but rather simply allocated into new cells, with replication beginning around the blastocyst stage in vertebrate species examined [28]. In *C. elegans*, Tsang and Lemire first showed that mtDNA CN increased substantially in development [12]. They also showed that when mtDNA replication was halted using the mtDNA-selective replication inhibitor ethidium bromide, nematodes did not develop past L3 [12]. Trifunovic and colleagues [13] later showed that loss of the sole replicative DNA polymerase also blocked development to adulthood, and Leung, Bess and colleagues found that mtDNA damage slowed larval development [14, 29, 30]. The fact that mtDNA CN either decreases slightly [12], stays constant [13] or increases only slightly [29] from the embryo to the L1 stage shows that a significant bottleneck must also occur in this species, since nuclear DNA (ncDNA) CN increases ~500-fold during this time. All these reports described a general increase in mtDNA content and requirement for mtDNA function

during development, with quantitative discrepancies that might be explained by methodological differences.

Similarly, in general, mitochondrial bioenergetics and metabolism change throughout development [31]. Characterizations of *C. elegans* aerobic metabolism during larval development have yielded somewhat conflicting results. Vanfleteren and De Vreese [32] reported peak protein-normalized OCR at the L3 and L4 stages, but Houthoofd et al. reported peak basal respiration at the L2 stage, after normalization to protein content [33]. Huang and Lin [34] developed a novel microfluidic device that permitted measurement of OCR on individual nematodes during development except for the L1 stage, at which OCR was not detectable with their methodology. Using this device, they reported steadily increasing OCR from L1 to L4 stages, as measured on a per-worm basis. They were also able to measure some specific aspects of OCR (e.g., ATP-linked, non-mitochondrial OCR, proton leak, and spare respiratory capacity), but reported these only at the L3, L4, and adult stages. They highlighted low ATP-linked OCR (on a per-worm normalization basis) at the L3 stage, and suggested that this corresponded to largely glycolytic energy production. Overall, the existing literature offers some insight into how mitochondrial bioenergetics change through *C. elegans* development. However, discrepancies exist, and a full time-course normalized to mitochondrial amount and incorporating analysis of the different components of oxygen consumption is lacking.

The goal of this study was to provide a comprehensive description of mitochondrial and non-mitochondrial OCR throughout larval development in *C. elegans*. We used a Seahorse XFe24 Extracellular Flux Analyzer to analyze whole animal basal OCR, basal mitochondrial OCR, ATP-linked OCR, spare respiratory capacity, and non-mitochondrial OCR, after optimizing the conditions for each of these at each larval stage. There are many ways in which OCR data can be normalized, each with its own strengths and limitations; we present and discuss results as normalized per individual worm, protein content, worm volume, mtDNA CN, ncDNA CN, and mtDNA:ncDNA ratio. Our results demonstrate when normalized to total protein, worm volume, or ncDNA or mtDNA count, after a 2-3-fold increase from L1 to L2 stages, changes in most OCR parameters from the L2 to the L4 stage are relatively modest. This indicates an earlier cellular shift to oxidative metabolism than suggested in most previous literature.

## 2. Methods

### 2.1 Synchronization of nematodes

We used the wildtype strain N2 Bristol and temperature-sensitive germline proliferation mutant *glp-1* strain JK1107 (q224), both of which were obtained from *Caenorhabditis* Genetics Center, which is funded by NIH Office of Research Infrastructure Programs (P40 OD010440). Worms were prepared using the hypochlorite bleach method of egg preparation [35]. A plate of mixed-stage worms was washed with K-medium (50 mM NaCl, 30 mM KCl, 10 mM NaOAc; pH 5.5) (K-med) solution and transferred to a 15mL tube. The worms were then washed twice with 15 mL of K-medium, and gravid adult worms were allowed to settle by gravity. We isolated the adults by aspirating the supernatant, then isolated eggs by treating gravid adults with hypochlorite bleach (1 mL 5N hypochlorite bleach, 500 μL 1M NaOH, and 3.5 mL K-medium). The 15mL tube containing the adult worms in bleach was placed in a 20 ˚C incubator on a shaker for 8 minutes. We briefly removed tubes to vortex and observed changes under the microscope every two minutes. When only fragments of worms could be seen, we increased the volume to 15mL with K-medium solution to stop the reaction and washed the eggs with K-medium twice, centrifuging between washes. Next, we placed 10μL of the egg solution in three drops on a slide and counted the number of eggs in 10 μL. We

transferred eggs to OP50-seeded K-agar plates [36] supplemented with 6.75 μM (final in-agar concentration) nystatin and incubated at 20˚C to allow development. We plated about 1000 eggs per plate for experiments with L1s; about 500 for experiments with L2s and L3s; and about 300 for experiments with L4s to avoid food deprivation during development. We removed plates from the incubator when worms reached the desired larval stage, which corresponded to 21h for mid-L1s, 34h for mid-L2s, 43h for mid-L3s, 46h for late-L3s, and 53h for mid-L4s.

## 2.2 Seahorse XFe24 Extracellular Flux Analyzer-based measurements of oxygen consumption rate

**2.2.1 Preparation and counting of nematodes.** Upon reaching the desired larval stage, we harvested nematodes by washing the OP50-seeded K-agar plates with K-medium into 15mL centrifuge tubes. Tubes were centrifuged for 30 seconds at 2200 RCF if worms were L1-L3s; we let L4s settle by gravity. We resuspended worms in the centrifuged tubes with K-medium and placed them in an orbital shaker for 20 minutes at 20˚C to allow gut clearing. Tubes were then centrifuged for 30 seconds at 2200 RCF (regardless of larval stage) and re-suspended in K-medium. To estimate the number of worms per microliter, we suspended the worms and transferred 3–4 drops of 20μL of worms to a glass slide and counted under a microscope. We calculated the concentration of worms per microliter and the volume needed to achieve the desired concentration depending on the nematode's development stage, given 525 μL total volume added to each well. The desired number of worms (we tested 1000–3000 for L1s; 300–700 for L2s; 150–250 for L3s; and 75 for L4s; the ideal number yielding OCR rates of 200–400 pmol oxygen/minute for the basal reading had been previously optimized for the L4 stage but not earlier stages) was transferred to the 24-well Seahorse plate, and if needed the wells were completed with K-medium to reach a total volume of 525 μL. The Xfe96 was utilized for testing the efficacy of rotenone and antimycin A at the L4 larval stage following the same general method as described in S4 Fig. As such, 20–50 L4s were utilized per well with a total well volume of 100 μL.

**2.2.2 Preparation of Seahorse XFe24 and Xfe96 Extracellular Flux Analyzer and assay plates.** We hydrated the Seahorse sensor cartridge overnight using 1mL per well of the XF Calibrant solution at room temperature for the Xfe24 and 100 μL of XF Calibrant for the Xfe96. The Seahorse software was set up by identifying injection strategies along with final ETC inhibitor concentration, solvent and percent solvent used in each port. Configurations for evaluation with the Xfe24 were set to include 8 cycles of Basal Measurements, 1 mix cycle to oxygenate the micro-chamber, 3 wait cycles to allow the worms to settle, and a measuring cycle to 3 minutes [37, 38]. 8 measurement cycles were used for measures of FCCP response, 14 for DCCD response and 4 for sodium azide response. 75 μL of the desired concentrations of each ETC-inhibitor were transferred to the appropriate port (FCCP or DCCD to Port A and sodium azide to Port B, both dissolved in DMSO and stored at -80 C until use as described [38]). Once appropriate ETC inhibitors were added to injection ports, the cartridge was loaded onto the Seahorse XFe24 Extracellular Flux Analyzer for calibration. Upon calibration, the utility plate (with XF Calibrant solution) was replaced with a 24-well plate containing nematodes for the OCR measurement assay to begin. The assay should be run without using the heater and in a cool room such that internal temperatures are between 20 and 25 ˚C, a range within which *C. elegans* OCR does not detectably vary [39]. The data collection process took about three hours per plate. Analyses with the Xfe96 specifically aimed to examine the use of rotenone and antimycin A versus sodium azide, so configurations for evaluation were set to include 6 cycles prior to the first injection and 10 cycles after injection 1 followed by 10 cycles

after injection 2. The following final well concentrations were used: 1% DMSO as a Vehicle control, 25 μM rotenone in 1% DMSO, 36 μM Antimycin A in 1% DMSO, 25 μm Rotenone and 36 μM Antimycin A in 1% DMSO, 40 μM DCCD, and 10 mM sodium azide. The following injection strategies were used (Injection 1 + Injection 2): vehicle + vehicle, rotenone + vehicle, antimycin A + vehicle, rotenone and antimycin A + vehicle, vehicle + azide, rotenone and antimycin A + azide, DCCD + azide, and DCCD + rotenone and antimycin A.

Please note that for *C. elegans* OCR analyses, DCCD is used rather than oligomycin, which is used in cell culture. This is because oligomycin does not appear to penetrate the *C. elegans* cuticle [37]. However, DCCD is unfortunately less specific than oligomycin, meaning that some caution should used in interpreting results.

**2.2.3 OCR analysis.** After each equipment OCR run, raw graphs were analyzed and wells that presented serious technical problems were discarded from the calculations. Problems that resulted in exclusion were: drug injection did not cause any changes in OCR; negative OCR levels; non-linear oxygen consumption caused by oxygen depletion (which can be seen by analyzing the raw oxygen levels). Equations and measurement timepoints used to calculate each bioenergetic parameter are shown in Fig 1. For the assessment of the effect of rotenone and antimycin A versus sodium azide, the average of all pre-injection measurements but the first and last was utilized for basal quantification. For vehicle, the average of all but the first and last measurements after injection 1 were utilized. For all remaining groups, the average of the lowest three measurements after injection of the drug(s) of interest were utilized.

## 2.3 Mitochondrial and nuclear DNA copy number

We measured mtDNA and ncDNA copy numbers per nematode as described [40]. We transferred 6 individual worms using the platinum worm pick into PCR tubes containing 90μL of 1x worm lysis buffer and froze them in the -80˚C freezer. After at least 10 minutes, we thawed the samples, vortexed and spun them briefly. We placed them in a thermal cycler to heat to 65 ˚C for 1 hour, then 95˚C for 15 minutes, then hold at 8 ˚C. The worm lysate was used as template DNA. We prepared a standard curve by thawing an aliquot of mtDNA copy number standard curve plasmid (10,000,000 copies/μL) and diluted it to 32,000 copies/μL. We serially diluted it to 4,000 copies/μL. The standard curve contains 64,000, 32,000, 16,000, 8,000, 4,000 and 0 copies per well. We then added 2μL of the worm lysate, 2μL of the mtDNA specific primer pair (400nM), 12.5μL SYBR Green PCR Master Mix), and 8.5μL of water in each well of the 96-well PCR plate. We amplified the target DNA using the ABI 7300 Real Time PCR system at 50 ˚C for 2 minutes, 95˚C for 10 minutes, 40 cycles of 95˚C for 15 seconds. We selected the option to calculate a dissociation curve for each sample to ensure the presence of a single product. The ncDNA copy number was quantified using the same method, however using a nuclear DNA-specific primer and a ncDNA standard curve that was prepared using the lysate of adult *glp-1* mutant *C. elegans* grown at 25 ˚C. Because this strain does not have a germline when the worms are raised at 25 ˚C, all adults have the same number of somatic cells and nuclear genomes, allowing the creation of a standard curve [29]. The lysate was prepared in 40μL of nuclease-free water, yielding 784 copies/μL. We serially diluted the lysate 1:1 until reaching 24.5 copies/μL to construct the standard curve.

## 2.4 Total protein content

Aliquots (2–3 replicates per experiment) of worms (~2000 for L1s and L2s; ~1000 for L3s; and ~500 for L4s) obtained from same sample used for the OCR measurements were transferred to 1.5mL tubes and frozen at -80˚C for total protein extraction. Once thawed, they were centrifuged, and the supernatant was removed. 150 μL of 10% SDS was added to the nematode

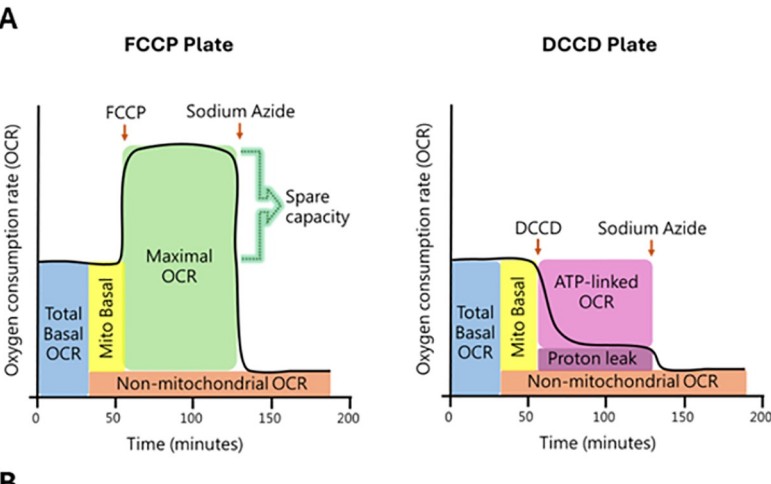

**Fig 1.** (A) Representative OCR curves showing the characteristic responses to the uncoupler FCCP and the mitochondrial inhibitors DCCD and sodium azide. Note that FCCP and DCCD injections are carried out separately (different plates or different wells) in *C. elegans*. (B) Equations and measurement timepoints used to calculate each bioenergetic parameter.

sample, and the samples were ultrasonicated (Model 3000 Ultrasonic Homogenizer, BioLogics, Inc.) for 2 cycles of 30s with the lowest amplitude setting. Complete nematode lysis was confirmed under the stereoscope. Protein content was measured using the Pierce™ BCA Protein Assay Kit (Thermo Scientific). Standard curve contained known concentrations of bovine serum albumin (1000, 750, 500, 250, 125, 125, 0 ug/ml) in the presence of 10% SDS.

## 2.5 Volume analysis

An aliquot of worms (100–200 individuals) obtained from same sample used for the OCR measurements was transferred to 6 cm K-agar plates lacking Bacto Peptone and allowed to air dry at 20 ˚C. Images were taken through automated scanning of the plate using the Keyence BZ-X700 microscope. Images were analyzed using the Worm Sizer plugin for ImageJ/Fiji [41].

Using this plugin, to minimize variation in measurement of worm volume, it is important to keep the image settings consistent between experiments, because worm volume is influenced by shadows cast onto the plate. This effect can be minimized by increasing the aperture stop.

## 2.6 Statistical analyses and graphical presentation

One-way ANOVA followed by Tukey's multiple comparisons test were applied using the ANOVA function in the car package in R version 4.2.1. Each experiment was repeated 2–4 times, with replicate wells on the same plate (which were aliquoted from the same group of cultured worms, which was counted as an independent biological replicate) counted as technical replicates. Error bars in graphs indicate standard errors of the mean, except in cases where n = 2, in which cases the bars indicate the two data points. Different letters on bars in a single graph indicate statistically significant differences; i.e., a bar with a given letter is statistically different from bars with different letters ($p < 0.05$ with Tukey's posthoc following ANOVA).

## 2.7 Data reporting

Data are available in S1 File and at the Duke University Research Data Repository https://doi.org/10.7924/r4hm5fv5t.

## 3. Results

### 3.1 Optimized conditions for OCR measurements in all larval stages

To establish the profile of *C. elegans* mitochondrial bioenergetics throughout different life stages we stage-synchronized wild-type (N2) nematodes (Fig 2A and 2B) and measured OCR levels with and without different drugs (electron transport chain inhibitors and a mitochondrial uncoupler). Optimization of worm number per well is critical to ensure that enough worms are present for reliable measurements (well above the background OCR and detection limits), but not so high that oxygen is depleted to the point of becoming limiting for cellular processes or for the reliability of the Seahorse Xfe24 Extracellular Flux Analyzer data (e.g. so high that re-oxygenation step is impaired). Drug optimization is critical both to ensure that a full response is observed, and to avoid a toxic response resulting in misleadingly decreased OCR. For example, insufficient uncoupling with FCCP will yield a falsely low estimate of maximal OCR and spare respiratory capacity. Too much uncoupling, however, will kill cells (and worms), leading to a decrease in OCR. Substrate depletion-mediated decreases in OCR may also be observed at later timepoints after FCCP injection; these timepoints should be excluded (although differences in time to depletion may be informative of substrate availability). Therefore, it is critical to optimize FCCP concentrations to show a maximal but steady OCR plateau but not a peak followed by a sharp decrease in OCR. We previously [16, 37, 42] reported optimization of number of worms per well and drug concentrations for a subset of larval stages, and here report optimized parameters for all larval stages.

For each larval stage, for optimization, we tested final well concentrations of 8, 25 or 75 μM for FCCP, and 20, 40 or 80 μM for DCCD. We initially tested 10, 20, 40, and 80 mM sodium azide, but found no additional inhibition of OCR and so used 10 mM used in all experiments shown here. We also tested 12.5, 50, and 90 μM FCCP in the L1 stage. Concentration optimization data for FCCP and DCCD are presented in S1 and S2 Figs. Using the criteria described above, we found that 25 μM FCCP was the best of the tested concentrations at all larval stages; 50 μM was also acceptable at L1 (but was not tested at other stages). 20 μM DCCD was consistently slow to achieve full inhibition of OCR, at all larval stages. 40 μM DCCD worked well at all larval stage except L4, where it was slow to take effect. 80 μM was best at L4, and may be

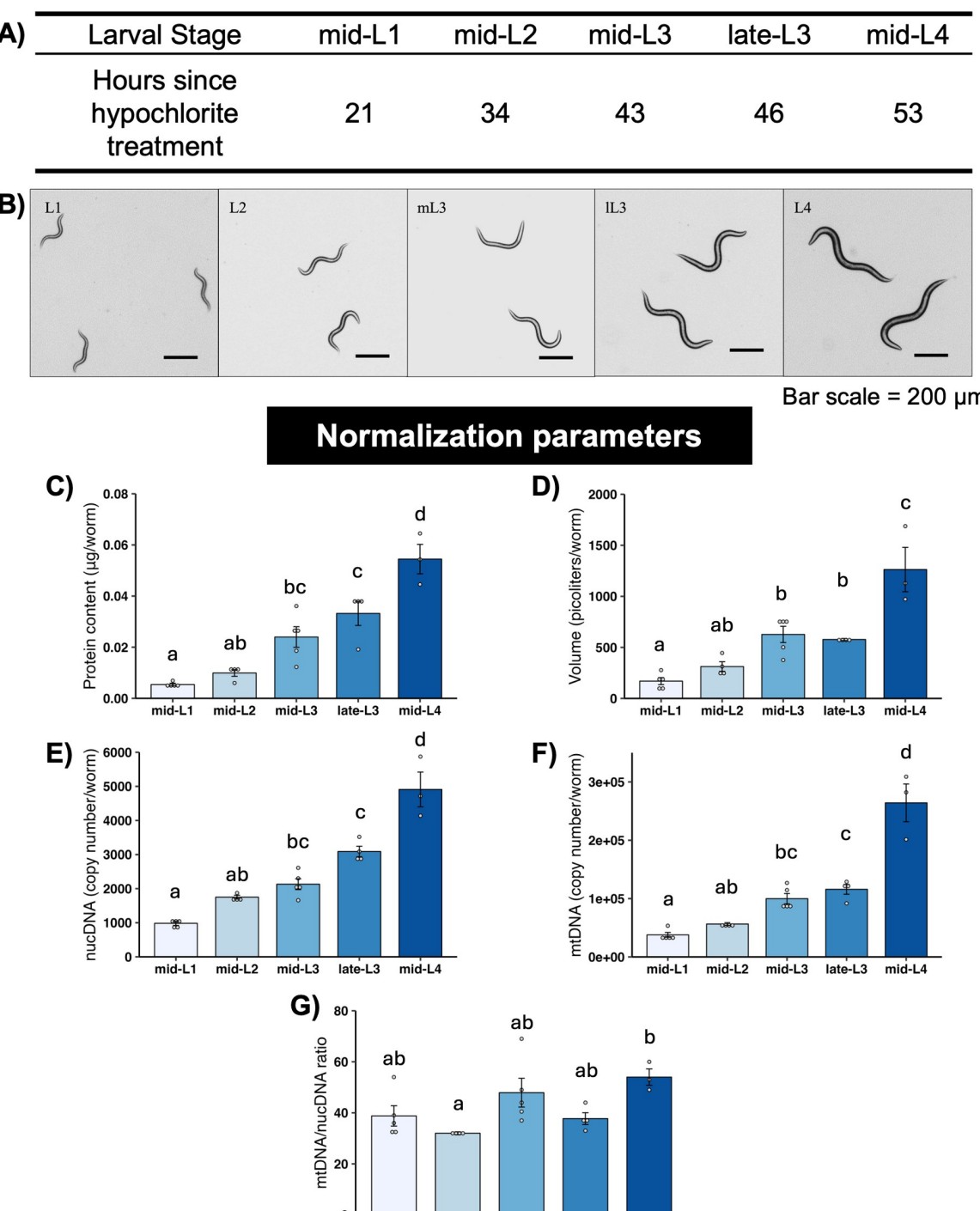

**Fig 2. Normalization parameters for different larval stages of *C. elegans*.** A) Larval stages and timepoints for each stage used in this study. B) Representative images of *C. elegans* at each stage. C) Total protein content at each stage. D) Worm volume at each stage. E) Nuclear DNA copy number at each stage. F) Mitochondrial DNA copy number at each stage. G) Mitochondrial/nuclear DNA ratio at each stage. All normalization endpoints had 3–5 biological replicates per stage. Different letters represent $p < 0.05$ with Tukey's posthoc following ANOVA.

acceptable at earlier larval stages, although in some cases 80 μM resulted in a puzzling waning of inhibition of OCR over time (S2 Fig). We note that while these concentrations were optimal among those we tested, under certain circumstances, by taking a subset of the top measurements after FCCP injection and lowest measurements after DCCD injection it may be possible to include data from a wider range of concentrations for comparisons between larval stages, increasing statistical power. For example, even in cases where FCCP or DCCD were slow to reach their full effect, by taking only the highest three and lowest two OCR values (respectively), it would be possible to measure FCCP-stimulated and DCCD-inhibited OCR accurately. Our most current, detailed protocol for measuring OCR using a Seahorse instrument is presented in S2 File. Note that full descriptions are provided for a 24-well instrument (used in this study), with some additional description relevant to using a 96-well instrument. Similarly, to maximize the utility of this protocol, we also include conditions for adult stages although we did not include adult data in this manuscript. Finally, we tested whether rotenone and antimycin A, often used together to inhibit all mitochondrial OCR (in place of the sodium azide used here), could be used in *C. elegans*. A previous report showed that 10 μM of each was ineffective during the timecourse of a typical Seahorse assay [39]. We found that inhibition of OCR was either too slow or incomplete to be practical in typical Seahorse assays. We evaluated the maximum soluble concentrations of 25 μM rotenone and 36 μM antimycin A in 1% DMSO in isolation and combination, and did not detect a decrease in OCR within two hours of injection (S3 Fig). The use of rotenone and antimycin A is further hindered by the requirement of DMSO for dissolution as the final well concentration of DMSO cannot exceed 2% but both DCCD and rotenone and Antimycin A require DMSO to maintain solubility.

## 3.2 Use of parallel DCCD and FCCP injections

Because we previously found that FCCP injection altered the response to subsequent sodium azide injection at the L4 stage [37], we tested whether this would be true at other larval stages. Indeed, FCCP decreased the degree to which sodium azide would inhibit OCR at all larval stages except late L3 (S4 Fig). We also found smaller effects of DCCD and DMSO on the response to sodium azide injection (S4 Fig). Therefore, for the experiments in this manuscript, we injected DCCD and FCCP in parallel, using separate plates, rather than in sequence and on the same plate, as is more typically done for cell culture analyses. As a result, for each larval stage measurement, we had separate measurements for basal OCR (i.e., OCR measured prior to injection of either DCCD or FCCP). We tested statistically for whether there was a plate effect for the basal OCR measurements, and as expected, found none; therefore, we combined those results. However, we used plate-specific measurements for calculation of maximal OCR and spare capacity (FCCP plates) and ATP-linked and proton leak-associated OCR (DCCD plates). We calculated non-mitochondrial and mitochondrial basal OCR values using DCCD plates.

## 3.3 mtDNA count, but not mtDNA:ncDNA ratio, increases from mid-L1 to mid-L4 stages

The measured larval stage-specific parameters other than worm count that we used to normalize OCR results are shown in Fig 2C–2G: total protein content, worm volume, mtDNA and ncDNA CN, and mtDNA:ncDNA ratio. Interestingly, mtDNA:ncDNA ratio only clearly increased at the L4 stage, and this increase was relatively modest (~20%). An important aspect of *C. elegans* development that informs interpretation of these values for normalization purposes is that somatic cell divisions are invariant in all worms, and germ cell division limited

through the L4 stage, such that ncDNA CN is essentially a readout for developmental progression. The numerical values of the parameters presented in Fig 2 are available in S1 Table.

### 3.4 Basal OCR

Basal OCR is simply the rate of total oxygen consumption measured in liquid without any additional injections or manipulations. It is generally comparable to the values obtained using older techniques such as Clark electrode-based measurements. Basal OCR increased consistently throughout development when results were normalized per worm or mtDNA/ncDNA ratio (Fig 3). In contrast, when basal OCR is normalized to volume, total protein, ncDNA CN, or mtDNA CN, there is a 2- to 3-fold increase from L1 to L2, but little or no increase throughout the later developmental stages.

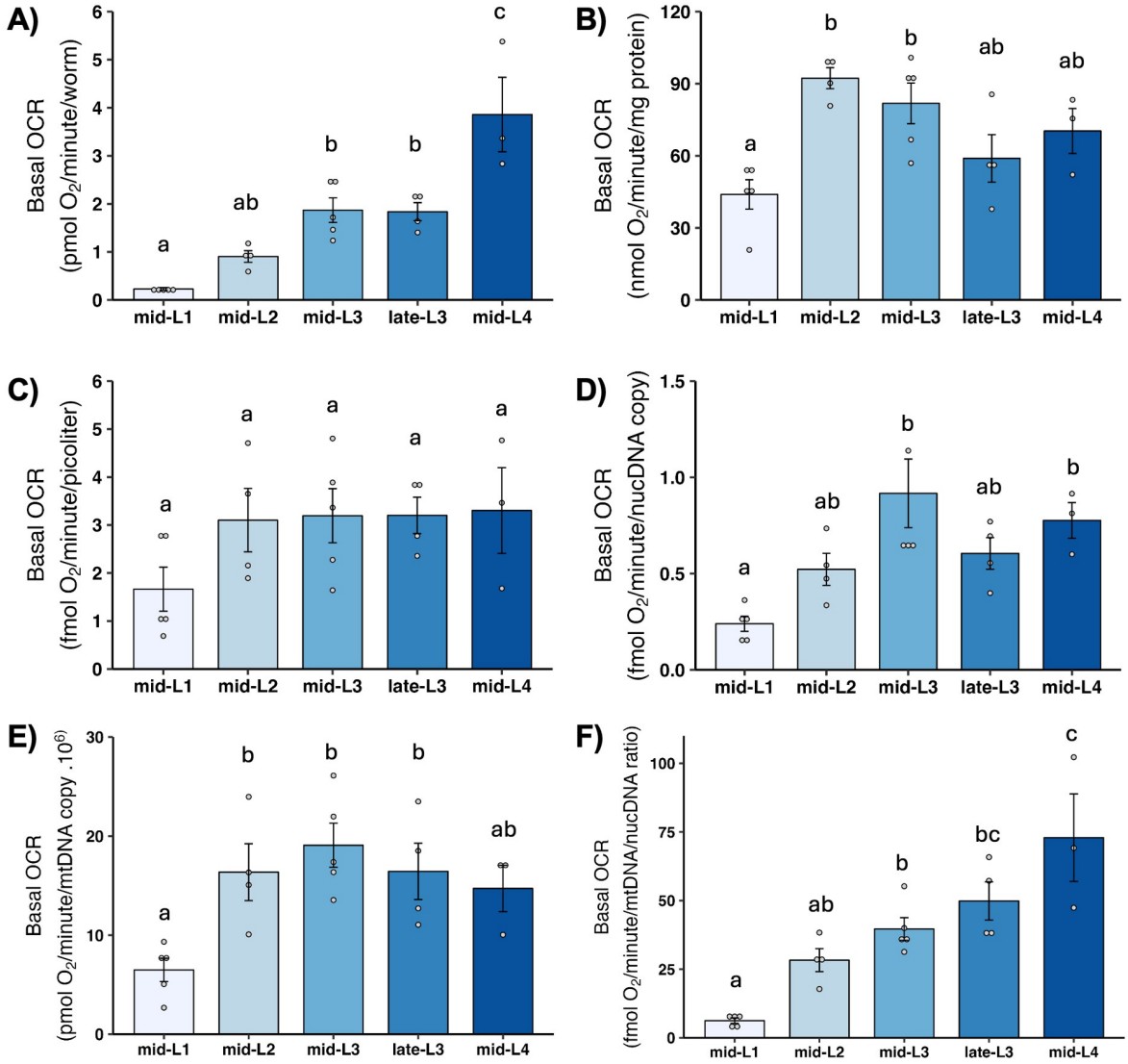

**Fig 3. Basal oxygen consumption rate (OCR) at different larval stages after normalization.** A) Basal OCR per worm. B) Basal OCR per mg protein. C) Basal OCR per volume. D) Basal OCR per ncDNA copy number. E) Basal OCR per mtDNA copy number. F) Basal OCR per mtDNA/ncDNA ratio. n = 3–5 biological replicates across stages. Different letters represent p < 0.05 with Tukey's posthoc following ANOVA.

### 3.5 Basal mitochondrial OCR

Basal mitochondrial OCR is that portion of basal OCR that can be attributed to mitochondrial oxygen consumption (i.e., this excludes non-mitochondrial OCR). Although large variance resulted in a lack of statistically significant differences, the general pattern of mean mitochondrial OCR was essentially the same as observed for basal OCR: substantial increases across larval stages disappear, with the exception of an increase from L1 to L2, upon normalization to volume, total protein, ncDNA CN, or mtDNA CN (Fig 4). This is highlighted by our calculation of the percent of total basal OCR that was mitochondrial: the percent was similar at all larval stages, 75–80% (Fig 4B).

### 3.6 ATP-linked OCR

ATP-linked OCR is defined experimentally as the amount of basal OCR that can be eliminated by inhibiting ATP synthase, and is intended to reflect the portion of mitochondrial oxygen consumption used to convert ADP to ATP—i.e., to "make energy." Developmental changes in ATP-linked OCR were similar to those observed for total mitochondrial OCR (Fig 5).

### 3.7 Maximal OCR and spare respiratory capacity

Maximal OCR is determined by chemically uncoupling mitochondria; FCCP shuttles protons back into the matrix without generating ATP, which causes mitochondria to consume oxygen faster in order to maintain the proton gradient. Thus, maximal OCR serves as a measure of how quickly mitochondria can consume oxygen. The increase in OCR from the basal level is termed "spare respiratory capacity," and reflects the ability of mitochondria to increase oxygen consumption on demand, which is important for stress response. We found a large increase in maximal OCR (Fig 6) and spare respiratory capacity (Fig 7) on per-worm basis, and a significant increase from L1 to L2 after other normalizations. A further increase occurred in most cases in one or both L3 stages, with a subsequent decrease in the L4 stage, although this was statistically significant only after normalization to mtDNA CN.

### 3.8 Proton leak

"Proton leak" refers to oxygen consumed by mitochondria when ATP synthase is blocked. Because oxygen can only be consumed by mitochondria when there is electron flow, and electron flow is blocked if there is no dissipation of the proton gradient, any mitochondrial oxygen consumption that occurs when ATP synthase is blocked must reflect dissipation of the proton gradient by "leak." "Leak" is thus defined as any such dissipation that is not mediated by ATP synthase, and includes important biological processes that may vary with developmental stage such as protein import, ion exchange, phosphate transport, transhydrogenase activity, and more [43]. We observed a pattern of apparent increase from L1 to L2, possibly peaking at L3 and then decreasing slightly upon normalization to factors other than worm count; however, these changes were not statistically significant (Fig 8).

### 3.9 ATP-linked OCR and spare respiratory capacity as a function of total basal OCR

To test whether the proportion of total cellular OCR that is used by mitochondria changes as a function of developmental stage, we plotted these ratios (panels B in Figs 4–8). We observed no differences.

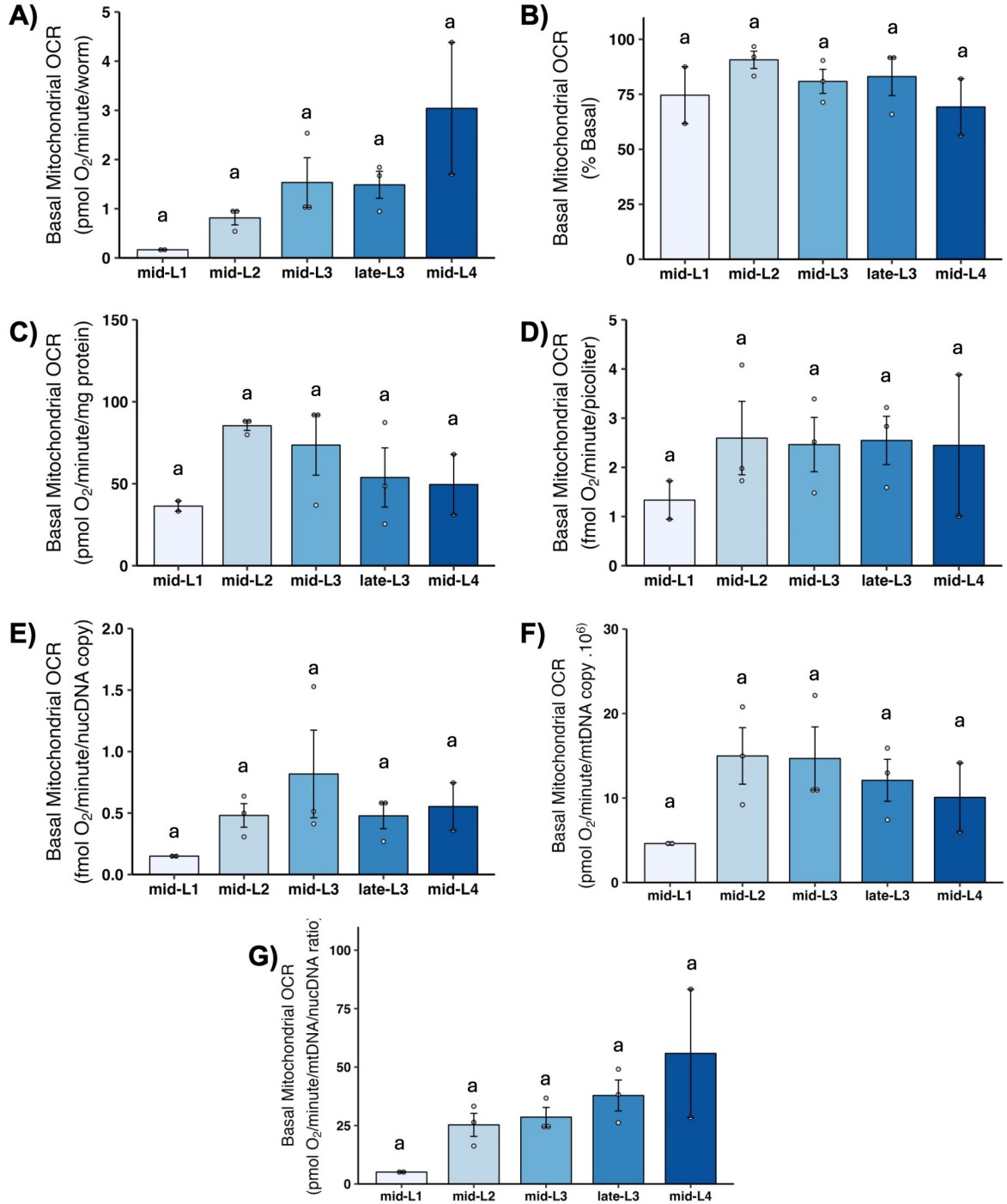

**Fig 4. Basal mitochondrial oxygen consumption rate (OCR) at different larval stages after normalization.** A) Basal mitochondrial OCR per worm. B) Basal mitochondrial OCR as percent total basal. C) Basal mitochondrial OCR per mg protein. D) Basal mitochondrial OCR per volume. E) Basal mitochondrial OCR per ncDNA copy number. F) Basal mitochondrial OCR per mtDNA copy number. G) Basal mitochondrial OCR per mtDNA/ncDNA ratio. n = 2–3 biological replicates across stages. Different letters represent p < 0.05 with Tukey's posthoc following ANOVA.

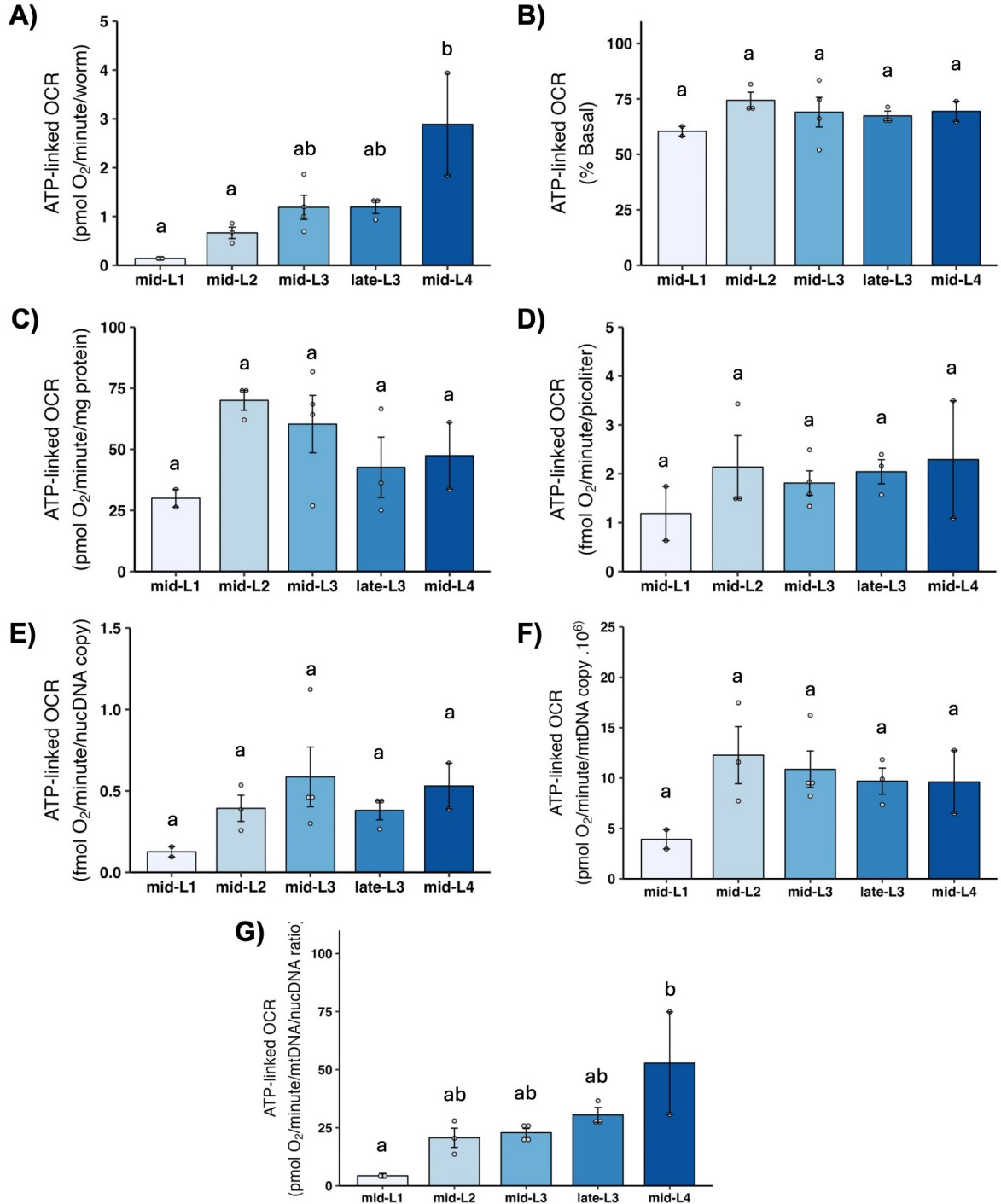

**Fig 5. ATP-linked mitochondrial oxygen consumption rate (OCR) at different larval stages after normalization.** A) ATP-linked OCR per worm. B) ATP-linked OCR as percent total basal. C) ATP-linked OCR per mg protein. D) ATP-linked OCR per volume. E) ATP-linked OCR per ncDNA copy number. F) ATP-linked OCR per mtDNA copy number. G) ATP-linked OCR per mtDNA/ncDNA ratio. Data shown using 40 μM DCCD for larval stages L1-late-L3 and 80 μM for L4 stage. n = 2–3 biological replicates across stages. Different letters represent p < 0.05 with Tukey's posthoc following ANOVA.

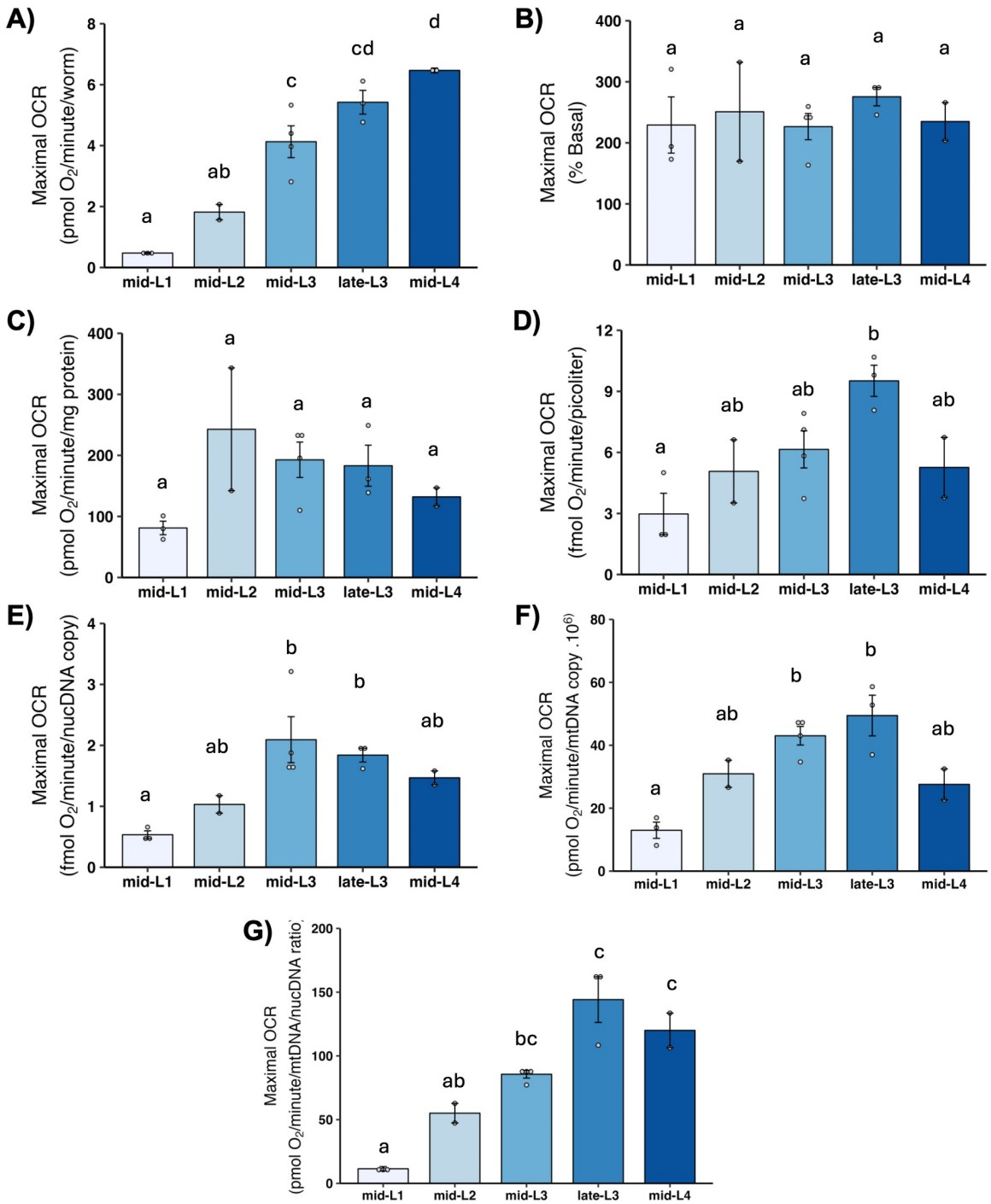

**Fig 6. Maximal oxygen consumption rate (OCR) at different larval stages after normalization.** A) Maximal OCR per worm. B) Maximal OCR as percent total basal. C) Maximal OCR per mg protein. D) Maximal OCR per volume. E) Maximal OCR per ncDNA copy number. F) Maximal OCR per mtDNA copy number. G) Maximal OCR per mtDNA/ncDNA ratio. Data shown using 25 μM FCCP across all stages. n = 2–3 biological replicates across stages. Different letters represent p < 0.05 with Tukey's posthoc following ANOVA.

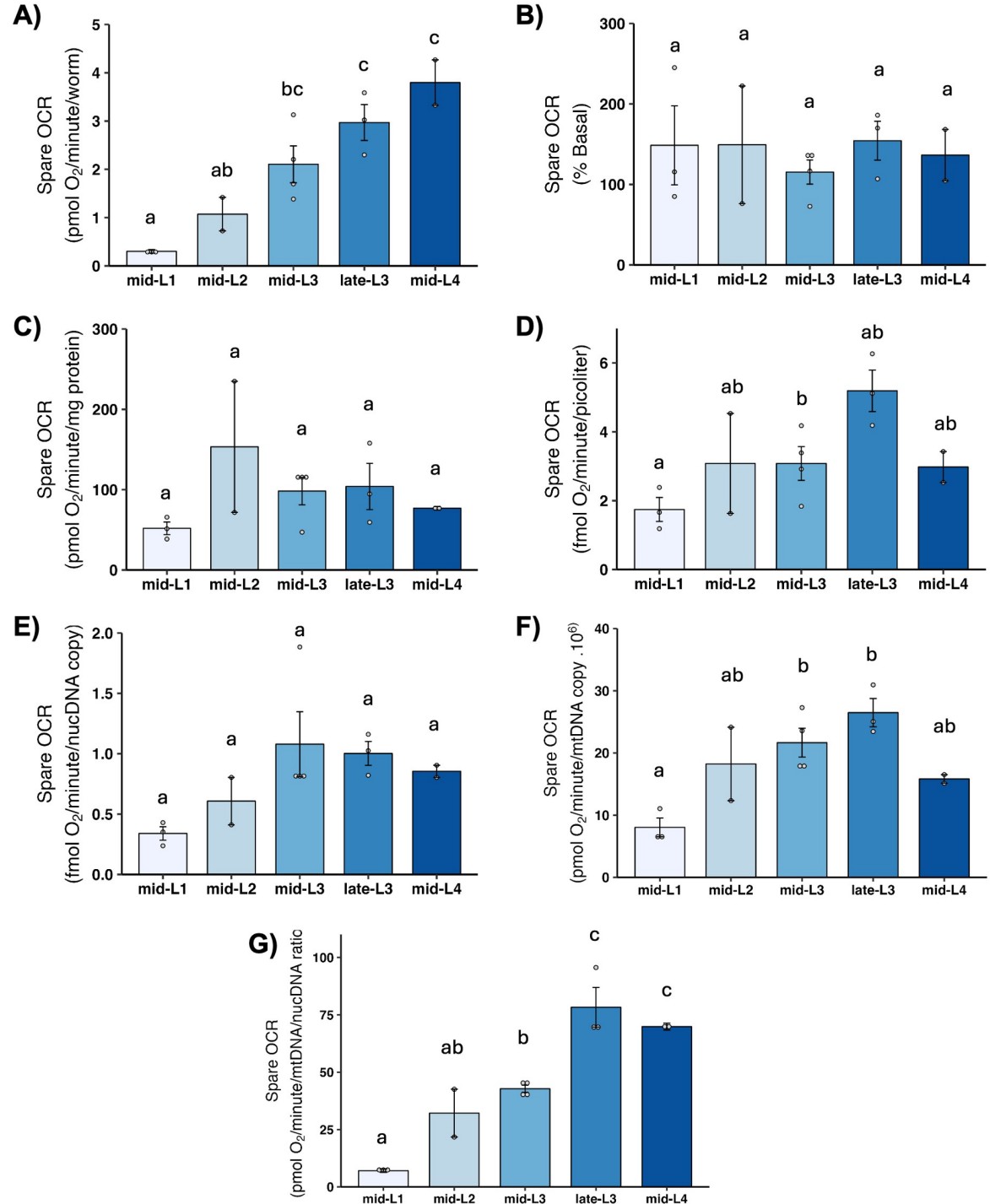

**Fig 7. Spare respiratory capacity at different larval stages after normalization.** A) Spare OCR per worm. B) Spare OCR as percent total basal. C) Spare OCR per mg protein. D) Spare OCR per volume. E) Spare OCR per ncDNA copy number. F) Spare OCR per mtDNA copy number. G) Spare OCR per mtDNA/ncDNA ratio. Data shown using 25 μM FCCP across all stages. n = 2–4 biological replicates across stages. Different letters represent p < 0.05 with Tukey's posthoc following ANOVA.

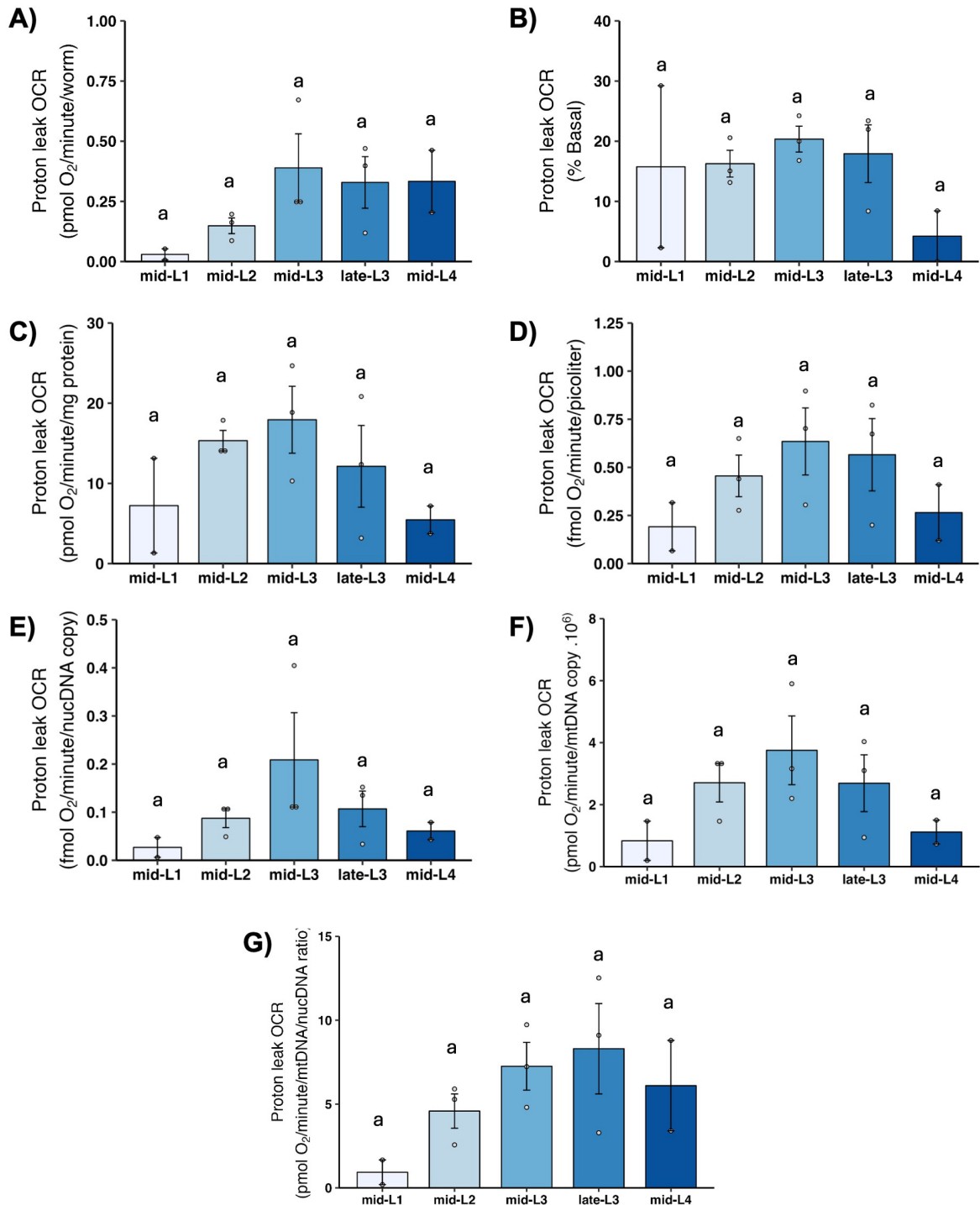

**Fig 8. Proton leak at different larval stages after normalization.** A) Proton leak OCR per worm. B) Proton leak OCR as percent total basal. C) Proton leak OCR per mg protein. D) Proton leak OCR per volume. E) Proton leak OCR per ncDNA copy number. F) Proton leak OCR per mtDNA copy number. G) Proton leak OCR per mtDNA/ncDNA ratio. n = 2–3 biological replicates across stages. Different letters represent p < 0.05 with Tukey's posthoc following ANOVA.

### 3.10 Non-mitochondrial OCR

Cellular processes other than the electron transport chain also consume oxygen; these include cytochrome P450 and other monooxygenases, NADPH oxidases, and other enzymes. Non-mitochondrial OCR is measured by inhibiting all oxidative phosphorylation using sodium azide, a potent complex IV/V inhibitor. Non-mitochondrial OCR increased substantially on a per-worm basis throughout development, but showed no change throughout development upon normalization by most other factors (Fig 9). In particular, the pattern of an increase in mitochondrial OCR-related parameters from L1 to L2 was not observed for non-mitochondrial OCR.

## 4. Discussion and conclusions

We report an optimized protocol for Seahorse-based analysis of OCR at all developmental stages in *C. elegans*. In the course of testing these parameters, we made two observations that are of biological interest. First, we learned that developmental trends in all measured OCR parameters vary qualitatively and quantitatively depending on how those values are normalized. Second, contrary to earlier ideas that most bioenergetic processes are anaerobic in early larval stages, we learned that there is significant oxidative phosphorylation occurring at all larval stages on a per-volume, per-protein, or per-mtDNA basis. Most of the normalized increase in oxidative phosphorylation that happens between the L1 and L4 stages occurs between L1 and L2, with a smaller increase observed in some parameters with some normalizations at the L3 stage. Basal OCR, maximal OCR, and spare respiratory capacity were slightly lower at the L4 than L3 stages upon normalization to mtDNA CN.

Very early developmental stages are reported to utilize energetic processes other than oxidative phosphorylation [44]. If and when this switch might occur in *C. elegans* has not been clearly defined. However, based on indirect evidence derived from a variety of phenotypes in which larval development is blocked or at least slowed when mtDNA replication, mitochondrial protein translation, or oxidative phosphorylation are inhibited, previous literature has suggested that the L3/L4 transition is accompanied by a metabolic switch from glycolysis to oxidative phosphorylation, or at least that inhibition of these processes signals for developmental delay or arrest at these (but not prior) stages [11–13, 29, 34]. This appeared consistent with a number of reports of dramatic increases in OCR through development when calculated on a per-nematode basis, as our results also show. Furthermore, Vanfleteren and De Vreese [32] reported peak protein-normalized OCR at the L3 and L4 stages, again consistent with a required metabolic switch at this stage. On the other hand, Houthoofd et al. reported peak basal respiration at the L2 stage, after normalization to protein content [33]. Our results clearly show significant oxidative phosphorylation at L1, a strong increase at L2, and possibly some additional increase at L3, depending on how normalization is done. These results suggest that however worms sense mitochondrial dysfunction and therefore arrest at L3, it is unlikely to be failure to activate oxidative phosphorylation, as any such inhibition should be able to be sensed by L2 or even L1 stage worms. One caveat to this conclusion is that our measurements were made at the whole-organism level and therefore cannot exclude a cell-specific oxidative phosphorylation-sensing mechanism. A second caveat is that it is possible that constraint of maximal (but not basal mitochondrial or ATP-linked) OCR could somehow be limiting for transition to L4. Such a constraint could be effective at the L3 to L4 stage because the maximal OCR is somewhat reduced in L4 compared to L3, in particular when normalized to mtDNA, which we assume serves as a proxy for amount of mitochondria. Another biological aspect of the L3 to L4 transition that could relate to developmental arrest is the onset of gonadal development and gametogenesis [45]. Cellular reactive oxygen species (ROS) and redox status are

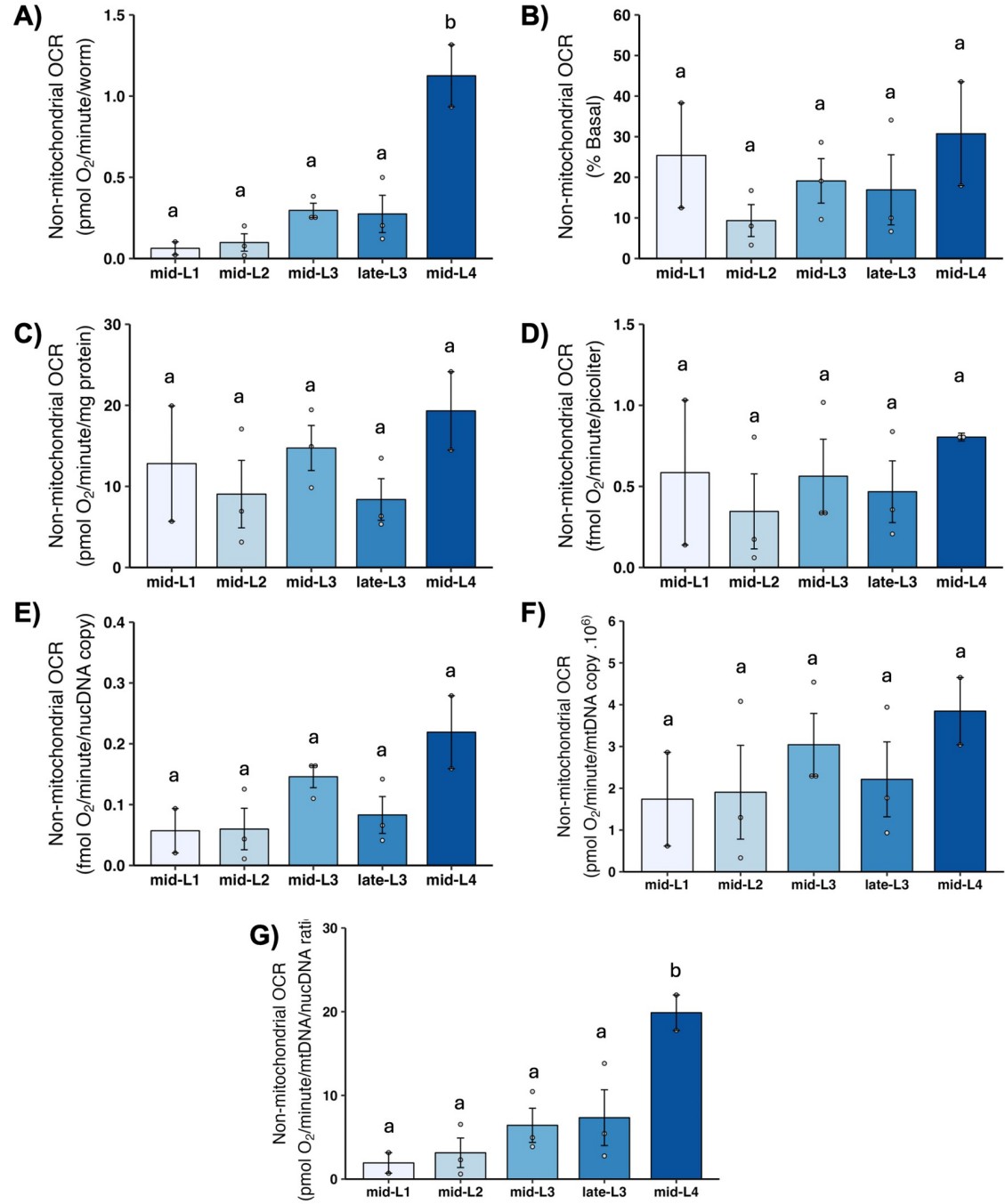

**Fig 9. Non-mitochondrial oxygen consumption rate (OCR) at different larval stages after normalization.** A) Non-mitochondrial OCR per worm. B) Non-mitochondrial OCR as percent total basal. C) Non-mitochondrial OCR per mg protein. D) Non-mitochondrial OCR per volume. E) Non-mitochondrial OCR per ncDNA copy number. F) Non-mitochondrial OCR per mtDNA copy number. G) Non-mitochondrial OCR per mtDNA/ncDNA ratio. n = 2–3 biological replicates across stages. Letters represent p < 0.05 with Tukey's posthoc following ANOVA.

essential for proliferation, differentiation, and migration, and mitochondrial ROS production has a major role in the control of such processes [46]. It is possible that the L3 arrest upon mitochondrial dysfunction may be due to a strict control of the redox status necessary for proper gametogenesis. Worms have a more reducing environment during the L3 to L4 transition [47] and dysfunctional mitochondrial may produce higher levels of ROS [48]. Thus, in the light of such findings, our results suggests that the L3 arrest upon mitochondrial dysfunction could result from signaling events or the necessity of a strict redox status control rather than a reliance of oxidative phosphorylation for energy production starting only at this developmental stage. We recommend future studies focusing on a deeper understanding of the mitochondrial control of *C. elegans* cellular redox status and development.

Choice of normalization parameters is critical. One approach is to normalize OCR to volume rather than number of worms. This is important even when analyzing OCR in cell culture [49], where cell size varies with cell cycle but to a much smaller degree than is the case for developing worms. From just-hatched L1s to late-L4 worms, there is nearly a 10-fold increase in volume [50] (our own results show a slightly smaller change, likely because we examined a slightly reduced developmental time course). Thus, worm number is not a useful normalization value unless all worms in the experiment are nearly identical in size. Note that even within larval stages there is significant growth [50], as is also true in adulthood. A second normalization approach is to protein content, either total or mitochondrial. This is helpful, although more work-intensive. A third option is to normalize to ncDNA CN, which serves as a proxy for cell number and developmental stage at least during development in *C. elegans*, since somatic cell development is identical in every individual in this species. This approach also fails to take into account cell size variation, however. It also works less well in gravid adults, where a high proportion of nuclear genomes comes from the germline, such that effects of experimental manipulations on OCR could be the indirect result of how those manipulations affect germ cell division rather than how they affect mitochondria function *per se*. Normalizing OCR to volume or ncDNA CN has the disadvantage of not demonstrating whether a change in OCR is a function of more or fewer mitochondria per worm or per cell, versus a function of the same number of mitochondria not consuming the same amount of oxygen as controls (e.g., because of a genetic deficiency or chemical effect). For this purpose, we recommend measuring mtDNA CN, although it is also possible (but more challenging) to measure total mitochondrial protein, or specific mitochondrial proteins, or other proxies for mitochondrial "amount." Such an approach allowed us to show that in worms, exercise improved mitochondrial function as measured by OCR and other parameters, without altering mtDNA CN, suggesting improved mitochondrial function on a per-mitochondrion basis [26]. Ultimately, which of these is chosen will depend on the experimental questions being asked; often, it will be valuable to compare the result of multiple normalization procedures. Of note, these considerations are also important in circumstances outside those examined in this study. For example, mtDNA CN is also regulated by age in adults [51] and starvation [42], and so changes in OCR in those contexts need to be interpreted in the context of changes in mitochondrial content.

In conclusion, we present a carefully optimized protocol for Seahorse-based analysis of oxygen consumption rate (OCR) throughout the developmental stages of *C. elegans*, and highlight the importance of careful parameter selection for normalization in such analyses. Moreover, our study enhances our understanding of *C. elegans* physiology throughout development and provides valuable insights applicable to broader contexts, including significant MRC-mediated oxygen consumption in all larval stages, especially after L1. Future work measuring gamete and embryonic metabolism, changes with age, and cell type-specific metabolism will further

increase our understanding of how mitochondrial metabolism contributes to worm development, physiological functions, stress response, and aging.

## Supporting information

**S1 File. Excel file containing all Seahorse, DNA copy number, volume, and protein analysis data.**
(XLSX)

**S2 File. Word document with our laboratory Seahorse protocol.** Please note that this protocol includes information for adult lifestages as well and larval lifestages, and information about using a 96- instead of 24-well Seahorse instrument.
(DOCX)

**S1 Table. Table containing the numerical values of the parameters presented in Fig 1.**
(TIF)

**S1 Fig.** S2 Fig shows oxygen consumption rates at the L1 (panel A), L2 (panel B), mid-L3 (panel C), late L3 (panel D), and L4 (panel E) stages at different timepoints after the injection of different concentrations of FCCP, normalized per worm. Blue line represents local polynomial regression fitting of the data for visualization with 95% confidence interval using the geom_smooth function in ggplot2 package in R version 4.2.1. Figures A-E represent data across 1–4 biological replicates with 3–6 technical replicates (L1–1–3 biological reps with 4–5 technical replicates; L2 –three biological reps with 3–5 technical replicates; L3 –four biological reps with 3–5 technical replicates; Late-L3- three biological reps 3–5 technical replicates; L4 –two biological reps with 3–5 technical replicates).
(TIF)

**S2 Fig.** S3 Fig shows oxygen consumption rates at the L1 (panel A), L2 (panel B), mid-L3 (panel C), late L3 (panel D), and L4 (panel E) stages at different timepoints after the injection of different concentrations of DCCD, normalized per worm. Blue line represents local polynomial regression fitting of the data for visualization with 95% confidence interval using the geom_smooth function in ggplot2 package in R version 4.2.1. Figures A-E represent data across 2–4 biological replicates with 3–6 technical replicates (L1 –two biological reps with 4–6 technical reps; L2 –three biological reps with 3–5 technical reps; L3 –four biological reps with 3–5 technical reps; Late-L3- three biological reps with 3–5 technical reps; L4 –two biological reps with 3–5 technical reps).
(TIF)

**S3 Fig. S3 Fig shows the effects of injection of rotenone and antimycin A on OCR.** For comparison to 10 mM sodium azide, maximum soluble doses of rotenone (25 μM) and antimycin A (36 μM) in 1% DMSO were examined alone, in combination, and after injection of 40 μM DCCD. Three biological replicates were performed utilizing L4 worms with 5–11 technical replicates per group. First, OCR per worm is shown throughout the course of the assay (A) and the calculated OCR corresponding to each treatment (B). To improve visual clarity and comparability, the results are also shown normalized to the treatment specific basal measurements occurring prior to the first injection (C, D), and the timepoint matched vehicle control (E, F). Statistical analysis was performed utilizing GraphPad Prism 10.2.3. A two-way ANOVA was used for each analysis, followed by Sidak's post-hoc for multiple comparisons with $p < 0.05$ as the cutoff for statistical significance. Comparisons are displayed by letter in which any treatment possessing the same letter is not statistically significantly different and only treatments

that have no identical letters are significantly different.
(TIF)

**S4 Fig. S4 Fig shows the effects of including DMSO solvent vs EPA water, prior FCCP injections, and prior DCCD injections on sodium azide-mediated inhibition of oxygen consumption rates at different larval stages.** Non-mitochondrial OCR normalized per worm and percent basal after injection with FCCP and DCCD. n = 1–4 biological replicates, p-values from two-way ANOVA.
(TIF)

## Acknowledgments

We thank Allie Gartland-Gray and Sophia Lafferty-Hess for assistance with depositing data to the Duke Data Repository.

## Author Contributions

**Conceptualization:** Danielle F. Mello, Christina M. Bergemann, Joel N. Meyer.

**Formal analysis:** Danielle F. Mello, Christina M. Bergemann, Joel N. Meyer.

**Funding acquisition:** Joel N. Meyer.

**Investigation:** Danielle F. Mello, Luiza Perez, Katherine S. Morton, Ian T. Ryde.

**Methodology:** Danielle F. Mello, Christina M. Bergemann, Katherine S. Morton, Joel N. Meyer.

**Project administration:** Joel N. Meyer.

**Resources:** Joel N. Meyer.

**Supervision:** Joel N. Meyer.

**Visualization:** Danielle F. Mello, Christina M. Bergemann, Katherine S. Morton.

**Writing – original draft:** Danielle F. Mello, Luiza Perez, Katherine S. Morton, Joel N. Meyer.

**Writing – review & editing:** Danielle F. Mello, Christina M. Bergemann, Joel N. Meyer.

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
