## [Decision Letter · Decision Letter 0]

16 Jul 2024

PONE-D-24-25658Comprehensive characterization of mitochondrial bioenergetics at different larval stages reveals novel insights about the developmental metabolism of Caenorhabditis elegansPLOS ONE

Dear Dr. Meyer,

Thank you for submitting your manuscript to PLOS ONE. After careful consideration, we feel that it has merit but does not fully meet PLOS ONE’s publication criteria as it currently stands. Therefore, we invite you to submit a revised version of the manuscript that addresses the points raised during the review process.

We look forward to receiving your revised manuscript.

Kind regards,

Xiaosheng Tan

Academic Editor

PLOS ONE

Journal Requirements:

"R01ES028218, P42ES010356, R01ES034270"

"This work was funded by the National Institute of Health (R01ES028218, P42ES010356, R01ES034270). Some strains were provided by the Caenorhabditis Genetics Center, which is funded by NIH Office of Research Infrastructure Programs (P40 OD010440)."

"R01ES028218, P42ES010356, R01ES034270"

5. We notice that your supplementary figures are uploaded with the file type 'Figure'. Please amend the file type to 'Supporting Information'.

**Additional Editor Comments:**

Please respond to each of the reviewer's comments individually and provide supplementary experiments as needed. We will reconsider the manuscript based on the detailed revisions and additions.

Reviewers' comments:

Reviewer's Responses to Questions

**Comments to the Author**

1. Is the manuscript technically sound, and do the data support the conclusions?

Reviewer #1: Yes

Reviewer #2: Yes

Reviewer #3: No

2. Has the statistical analysis been performed appropriately and rigorously? 

Reviewer #1: Yes

Reviewer #2: Yes

Reviewer #3: Yes

3. Have the authors made all data underlying the findings in their manuscript fully available?

Reviewer #1: Yes

Reviewer #2: Yes

Reviewer #3: Yes

4. Is the manuscript presented in an intelligible fashion and written in standard English?

Reviewer #1: Yes

Reviewer #2: Yes

Reviewer #3: Yes

5. Review Comments to the Author

Reviewer #1: The manuscript PONE-D-24-25658, entitled "Comprehensive characterization of mitochondrial bioenergetics at different larval stages reveals novel insights about the developmental metabolism of Caenorhabditis elegans by Mello D.F. et al., reports results on the functional activity of mitochondria during C. elegans development by using a SeaHorseXFe24. By combining OCR determination with quantitative measurements such as mtDNA and nuclear DNA, as well as the use of drugs targeting the mitochondrial electron transport chain, the manuscript highlights a previously undiscovered ROC change in oxidative metabolism during early C. elegans development. This manuscript shows an interesting analysis of mitochondrial function during C. elegans development that fills a gap in equivalent analyses using the same type of SeaHorse XFe24 measuring device.

Supplemental files are useful for a comprehensive analysis of the data described therein. A comprehensive reference list is provided.

I think this manuscript should be considered for publication in Plos One after considering the following aspects that need to be clarified.

With kind regards,

Reviewer

Major modifications

As described in the chapter 2.3 "Mitochondrial and nuclear DNA copy number", lysates of the Glp-1 mutant line were used to produce the standard curves.

At what temperature was this sterile Glp-1 mutant line grown? Thank you for clarifying this point.

Can the culture temperature of the Glp-1 sterile mutant line used to prepare the lysates for the standard curves be considered equivalent to that of the N2 nematodes used for the study and reared at 20°C as indicated in the materials and methods section? Please discuss and argue this point in the discussion and conclusion sections.

Supporting info section.

In this methods section, a precise description of the temperature at which respiration measurements were carried out in the XFe24 device is given, indicating between 20°C and 25°C. The usual temperature for maintaining Wild type lines of C. elegans in culture is 20°C, and genetic crosses are generally carried out at this same temperature.

A previous report (Koopman et al., 2016, Nature Protocols, 11, 1798-1816; doi:10.1038/nprot.2016.106) describing C. elegans respiration measurements in the XFe96 device indicates, in Figure 4, that C. elegans N2 wild type respiration levels between temperatures of 20°C and 25°C are comparable.

Please clarify this point with arguments to support the conclusion that a temperature variation of 5°C (i.e. 20-25°C) during respiration measurements does not significantly impact the measurements when using the XFe24 apparatus.

Minor modifications

Section 3.10 Non-mitochondrial OCR, at line 392, the sodium azide is believed as a complex IV/V inhibitor. Please replace “potent complex IV inhibitor” by “potent complex IV/V inhibitor”

Section Discussion and conclusions

at line 421, the authors state that "previous literature has suggested that the L3/L4 transition is accompanied by a metabolic shift from glycolysis to oxidative phosphorylation, ...". Please add at least one reference to support this assertion.

Strains

Glp-1 mutants are mentioned in chapter 2.3 Mitochondrial and nuclear DNA copy number. However, as for the N2 nematodes, no origin of the lines is mentioned.

In a first paragraph of the material and methods section, please provide the information needed to understand the stains used for the experimental measurements described in the manuscript. Please indicate clearly the Glp-1 temperature -sensitive mutant allele used and its origin, as well as the origin of the WT N2 line used in the manuscript, and, if applicable, clearly quote "Some strains were provided by the CGC, which is funded by NIH Office of Research Infrastructure Programs (P40 OD010440)" if these nematode lines were obtained from the CGC.

Reviewer #2: Danielle Mello et al. performed an extensive analysis of mitochondrial and non-mitochondrial oxygen consumption rates (OCR) across various larval developmental stages in C. elegans using Seahorse-based methodologies. Their findings, detailed protocol, and comparison of different normalization methods offer significant insights for researchers conducting similar experiments. I have some minor suggestions that could enhance the clarity of the paper.

1. The authors used bar graphs throughout the paper to summarize their results without showing the raw Seahorse graphs. Including one or more representative Seahorse assay graphs alongside Table 1 would enhance the reader's understanding of how the data analysis was conducted.

2. While the bar graphs are clearly presented, the method of denoting statistical significance using letters can be difficult to interpret. I recommend using asterisks to denote levels of statistical significance (e.g., *, **, ***). Alternatively, the authors should consider providing legends for the bar graphs to explain the meaning of each letter. Additionally, overlaying individual data points on the bar graphs would help illustrate the data distribution and variability.

3. The authors stated that they initially tested 20, 40, and 80 mM sodium azide but observed no additional inhibition of OCR, leading them to use 10 mM in all experiments presented in the paper. However, the data supporting this decision is not included, and the rationale for selecting 10 mM is not entirely clear. I recommend that the authors provide the supporting data and clarify the reasoning.

Reviewer #3: In the paper of Meyer et al., some critical points are raised in the Seahorse analysis.

The Authors use DCCD indeed of oligomycin to block the ATP synthase and obtain the ATP production. There are several critical points. DCCD is not a specific inhibitor of ATP synthase but, in general, bounds the protonable carboxylic groups of aspartate or glutamate in a pH-dependent mode. Moreover, the inhibition of proton translocation of the FO domain of ATP synthase, by stable dicyclohexyl-N-acyl urea with c subnits, is concentration-dependent. High concentration can interact with the catalytic domain of the F1 sector of the enzyme. It is important to verify the inhibitory effect of DCCD on ATP synthesis during ADP phosphorylation and perform a titration curve. In addition to this, why the authors do not share the OCR profile of the mitostress test?

In the mitostress test, cell respiration is inhibited by rotenone+ antimycin A. Could the Authors explain the substitution win KCN?

In Table 1 some parameters are wrong.

- I do not know the role or Total basal OCR. Can the Authors explain this?

- Basal respiration can be explained as the initial respiration without non-mitochondrial respiration

- Maximal OCR is the highest rate measurement after FCCP injection without Non-Mitochondrial OCR

- Spare capacity (Maximal OCR) – (Basal OCR)

- ATP-linked OCR is (Basal OCR) – (2 lowest rate measurements after oligomycin injection)

- Proton leak is (basal OCR) – (ATP-linked OCR)

In my opinion, only the inadequacy of the evaluation of the bioenergetic parameters of cellular metabolism are sufficient to invalidate the results of the work and therefore the data presented are not reliable.

6. PLOS authors have the option to publish the peer review history of their article (what does this mean?). If published, this will include your full peer review and any attached files.

Reviewer #1: No

Reviewer #2: No

Reviewer #3: No

---

## [Author Response · Author response to Decision Letter 0]

11 Sep 2024

Dear Editors, 

Thank you for the opportunity to submit a revised draft of our manuscript “Comprehensive characterization of mitochondrial bioenergetics at different larval stages reveals novel insights about the developmental metabolism of Caenorhabditis elegans.” We thank the reviewers for their careful consideration and suggestions for this paper. We believe that the incorporation of their feedback, and additional experiments, have made our manuscript stronger. Please see below, in blue text, our responses to each of the reviewer comments. All page numbers refer to the revised manuscript with tracked changes. We have also made a few other small changes, including a supplemental table with the numerical values for the graphs shown in Figure 1.

Reviewer #1: The manuscript PONE-D-24-25658, entitled "Comprehensive characterization of mitochondrial bioenergetics at different larval stages reveals novel insights about the developmental metabolism of Caenorhabditis elegans by Mello D.F. et al., reports results on the functional activity of mitochondria during C. elegans development by using a SeaHorseXFe24. By combining OCR determination with quantitative measurements such as mtDNA and nuclear DNA, as well as the use of drugs targeting the mitochondrial electron transport chain, the manuscript highlights a previously undiscovered ROC change in oxidative metabolism during early C. elegans development. This manuscript shows an interesting analysis of mitochondrial function during C. elegans development that fills a gap in equivalent analyses using the same type of SeaHorse XFe24 measuring device.

Supplemental files are useful for a comprehensive analysis of the data described therein. A comprehensive reference list is provided.

I think this manuscript should be considered for publication in Plos One after considering the following aspects that need to be clarified.

With kind regards,

Reviewer

Major modifications

As described in the chapter 2.3 "Mitochondrial and nuclear DNA copy number", lysates of the Glp-1 mutant line were used to produce the standard curves. At what temperature was this sterile Glp-1 mutant line grown? Thank you for clarifying this point.

Glp-1 mutants for this purpose were grown at 25 C in order to eliminate germline proliferation (this is a temperature-sensitive mutant). Because somatic cell number is invariant in adult C. elegans, this procedure results in precise knowledge of the number of cells and hence nuclear genomes, enabling construction of a standard curve. This technique was originally described in Leung et al., 2013 (which is already cited in the manuscript, in another context): https://pubmed.ncbi.nlm.nih.gov/23374645/. We have added additional text to explain this to the Methods (lines 224-229), including a reference to that original publication.

Can the culture temperature of the Glp-1 sterile mutant line used to prepare the lysates for the standard curves be considered equivalent to that of the N2 nematodes used for the study and reared at 20°C as indicated in the materials and methods section? Please discuss and argue this point in the discussion and conclusion sections.

Because this method was previously described, rather than repeat the full discussion here, we have chosen to simply reference the original Leung et al., 2013 paper in which this technique was used (lines 224-229). At the editor’s discretion, we can give more detail or quote the original paper:

“Nuclear copy number was determined by creating a standard curve for the nuclear DNA based on young adult (24 h post-L4) glp-1 mutant nematodes raised at 25°C. At this temperature, this strain has a fixed number of cells since it has no germline proliferation [30] and C. elegans somatic cells do not divide in adulthood [31]. We based the standard curve on the calculation that adults lacking germ cell proliferation would contain 3134 genomic copies [32,33].”

31. Sulston J: Cell Lineage. In The Nematode Caenorhabditis elegans. Edited by Wood WB. Cold Spring Harbor, NY: Cold Spring Harbor Laboratory Press; 1988:123–155.

32. Golden TR, Beckman KB, Lee AH, Dudek N, Hubbard A, Samper E, Melov S: Dramatic age-related changes in nuclear and genome copy number in the nematode Caenorhabditis elegans. Aging Cell 2007, 6:179–188.

33. Emmons SW: The Genome. In The Nematode Caenorhabditis elegans. Edited by Wood WB. Cold Spring Harbor, NY: Cold Spring Harbor Laboratory Press; 1988:47–79.

Supporting info section.

In this methods section, a precise description of the temperature at which respiration measurements were carried out in the XFe24 device is given, indicating between 20°C and 25°C. The usual temperature for maintaining Wild type lines of C. elegans in culture is 20°C, and genetic crosses are generally carried out at this same temperature.

A previous report (Koopman et al., 2016, Nature Protocols, 11, 1798-1816; doi:10.1038/nprot.2016.106) describing C. elegans respiration measurements in the XFe96 device indicates, in Figure 4, that C. elegans N2 wild type respiration levels between temperatures of 20°C and 25°C are comparable.

Please clarify this point with arguments to support the conclusion that a temperature variation of 5°C (i.e. 20-25°C) during respiration measurements does not significantly impact the measurements when using the XFe24 apparatus. We greatly appreciate the reviewer bringing this to our attention, and have now referenced this publication (both in the supporting information protocol and in the main manuscript, lines 177-178) to support the likelihood that any differences in OCR between 20 and 25 C are likely to be small. 

Minor modifications

Section 3.10 Non-mitochondrial OCR, at line 392, the sodium azide is believed as a complex IV/V inhibitor. Please replace “potent complex IV inhibitor” by “potent complex IV/V inhibitor”

Thank you, we have corrected this. 

Section Discussion and conclusions

at line 421, the authors state that "previous literature has suggested that the L3/L4 transition is accompanied by a metabolic shift from glycolysis to oxidative phosphorylation, ...". Please add at least one reference to support this assertion.

The references for this phrase are at the end of the same sentence: “previous literature has suggested that the L3/L4 transition is accompanied by a metabolic switch from glycolysis to oxidative phosphorylation, or at least that inhibition of these processes signals for developmental delay or arrest at these (but not prior) stages [11-13, 29, 34].” The specific findings reported in those references and supporting these ideas are discussed in the Introduction and subsequent parts of the Discussion.

Strains

Glp-1 mutants are mentioned in chapter 2.3 Mitochondrial and nuclear DNA copy number. However, as for the N2 nematodes, no origin of the lines is mentioned.

In a first paragraph of the material and methods section, please provide the information needed to understand the stains used for the experimental measurements described in the manuscript. Please indicate clearly the Glp-1 temperature -sensitive mutant allele used and its origin, as well as the origin of the WT N2 line used in the manuscript, and, if applicable, clearly quote "Some strains were provided by the CGC, which is funded by NIH Office of Research Infrastructure Programs (P40 OD010440)" if these nematode lines were obtained from the CGC.

Thank you, we have added this information (lines 120-122). We do note however that it is our understanding that PLOS policy does not permit inclusion of grant information in the manuscript, so the “which is funded by NIH Office of Research Infrastructure Programs (P40 OD010440)” part may need to be deleted.

Reviewer #2: Danielle Mello et al. performed an extensive analysis of mitochondrial and non-mitochondrial oxygen consumption rates (OCR) across various larval developmental stages in C. elegans using Seahorse-based methodologies. Their findings, detailed protocol, and comparison of different normalization methods offer significant insights for researchers conducting similar experiments. I have some minor suggestions that could enhance the clarity of the paper.

1. The authors used bar graphs throughout the paper to summarize their results without showing the raw Seahorse graphs. Including one or more representative Seahorse assay graphs alongside Table 1 would enhance the reader's understanding of how the data analysis was conducted. As suggested, we have added a new Figure 1. 

2. While the bar graphs are clearly presented, the method of denoting statistical significance using letters can be difficult to interpret. I recommend using asterisks to denote levels of statistical significance (e.g., *, **, ***). Alternatively, the authors should consider providing legends for the bar graphs to explain the meaning of each letter. We feel that letters are more helpful in this case because we want to compare each stage to every other stage, not all stages to one that is designated a “baseline.” For example, we don’t just want to know if each stage is different from L1; we also want to know if L4 is different from the L3 stages. We think that letters are better than asterisks for this purpose. To clarify interpretation, we added this sentence to the Methods paragraph on statistical analysis: “Different letters on bars in a single graph indicate statistically significant differences; i.e., a bar with a given letter is statistically different from bars with different letters (p < 0.05 with Tukey’s posthoc following ANOVA).” And the shorter sentence “Different letters represent p < 0.05 with Tukey’s posthoc following ANOVA.” is in all relevant Figure legends.

Additionally, overlaying individual data points on the bar graphs would help illustrate the data distribution and variability. 

We have added the individual data points (with technical replicates averaged).

3. The authors stated that they initially tested 20, 40, and 80 mM sodium azide but observed no additional inhibition of OCR, leading them to use 10 mM in all experiments presented in the paper. However, the data supporting this decision is not included, and the rationale for selecting 10 mM is not entirely clear. I recommend that the authors provide the supporting data and clarify the reasoning. 

Thank you for pointing out this discrepancy. Indeed, we mistakenly omitted to mention that we also tested the 10 mM sodium azide concentration, which we had previously used (Luz et al., 2015). We selected this concentration because we observed no further increase in OCR inhibition at higher concentrations. We have changed the text to clarify this point. The text now reads: "We initially tested 10, 20, 40, and 80 mM sodium azide, but found no dose-dependent increase in inhibition of OCR and so used 10 mM in all experiments shown here." 

Reviewer #3: In the paper of Meyer et al., some critical points are raised in the Seahorse analysis.

The Authors use DCCD indeed of oligomycin to block the ATP synthase and obtain the ATP production. There are several critical points. DCCD is not a specific inhibitor of ATP synthase but, in general, bounds the protonable carboxylic groups of aspartate or glutamate in a pH-dependent mode. Moreover, the inhibition of proton translocation of the FO domain of ATP synthase, by stable dicyclohexyl-N-acyl urea with c subnits, is concentration-dependent. High concentration can interact with the catalytic domain of the F1 sector of the enzyme. It is important to verify the inhibitory effect of DCCD on ATP synthesis during ADP phosphorylation and perform a titration curve. 

We agree that it would be preferable to use oligomycin, and indeed first tried to use oligomycin. Unfortunately, it appears that oligomycin does not penetrate the nematode cuticle within the timeframe of our assay, which is why we previously optimized this assay using the less specific FOF1 inhibitor DCCD; that process and the oligomycin data (including with a cuticle-deficient mutant) were published in Luz et al. PLOS ONE 2015 and Luz et al. 2015. Curr Protoc Toxicol. While we acknowledge this limitation and its impact on data interpretation, which should be considered, we believe this remains the most effective in vivo approach currently available for measuring ATP-linked OCR. We have added some text to highlight this reasoning and the potential lack of specificity (lines 188-191).

In addition to this, why the authors do not share the OCR profile of the mitostress test? 

As also requested by Reviewer 2, we have added a new Figure 1 showing the profiles. However, please note that the Figure we provide looks a bit different than the classical Mitostress Test. This is because for OCR analyses in C. elegans, it is not feasible to use all inhibitors in a single Seahorse run due to significant chemical interference, as detailed in section 3.2. Therefore, we present a schematic with two panels, one each for FCCP and DCCD injections. We note that all the OCR curves (but not full profiles) are presented within the supplementary files.

In the mitostress test, cell respiration is inhibited by rotenone+ antimycin A. Could the Authors explain the substitution with KCN? 

As noted, we use NaN3, sodium azide, in this work. Rotenone and antimycin A do inhibit Complexes I and III respectively in worms. However, they face two challenges for the use in Seahorse assays: speed and solubility. They appear to be taken up only slowly in worms, at least at the concentrations that can be achieved in the stock (injection) solutions given their solubility in DMSO. As a result, they inhibit OCR either too slowly or incompletely to be effective in a standard Seahorse assay. For example, Koopman et al., 2016: https://pubmed.ncbi.nlm.nih.gov/27583642/ reported a complete lack of effect of 10 uM rotenone or antimycin A in the timecourse of their Seahorse runs (Fig. S1). We now cite this in the main text. Also, we previously reported that a 1-hour incubation was required to reliably reduce ATP levels: https://pubmed.ncbi.nlm.nih.gov/27479364/ even with 80 uM of each. However, a detailed evaluation of the timecourse of OCR inhibition in a Seahorse at the maximal achievable concentrations has not to our knowledge been previously published, and so we carried out the experiments to provide this information, which we now include as Figure S3. The maximum achievable (soluble) well concentrations of 25 uM for rotenone and 36 uM for antimycin A did not detectably decrease OCR within 2 hours of injection separately or combination. We additionally provided the effect of sodium azide and each combination post DCCD injection to further clarify which is superior independently and after DCCD injection. Finally, due to their employment post DCCD injection, both DCCD and the rotenone and antimycin A combination can be dissolved in a maximum of 1% DMSO. As DCCD already presents a challenge of limited solubility, this makes the water soluble alternative sodium azide advantageous by allowing for up to 2% DMSO with DCCD. 

In Table 1 some parameters are wrong.

Most of these were correct, as explained below. We also hope that the new Figure 1 will clarify.

- I do not know the role or Total basal OCR. Can the Authors explain this? 

This parameter measures the total oxygen consumption by organisms, encompassing both mitochondrial and non-mitochondrial metabolism. Its broad scope makes it a valuable physiological marker, as it can be influenced by various test conditions and may be particularly relevant depending on the specific scientific question being investigated. It is also the parameter most similar to older Clark electrode measurements in whole worms (which were done in some of the literature we cited and compared out results to).

- Basal respiration can be explained as the initial respiration without non-mitochondrial respiration. 

We adopted the term "mitochondrial basal OCR" instead of the commonly used "basal respiration" because we believe it more accurately reflects the parameter being measured, and to distinguish clearly from total basal OCR, which as we explained in the preceding paragraph has its own utility.

- Maximal OCR is the highest rate measurement after FCCP injection without Non-Mitochondrial OCR. 

The OCR analyses in C. elegans differ significantly from the classical methods u

---

## [Decision Letter · Decision Letter 1]

20 Sep 2024

PONE-D-24-25658R1Comprehensive characterization of mitochondrial bioenergetics at different larval stages reveals novel insights about the developmental metabolism of Caenorhabditis elegansPLOS ONE

Dear Dr. Meyer,

Thank you for submitting your manuscript to PLOS ONE. After careful consideration, we feel that it has merit but does not fully meet PLOS ONE’s publication criteria as it currently stands. Therefore, we invite you to submit a revised version of the manuscript that addresses the points raised during the review process.

**Please provide a detailed explanation and response to the key issues raised by the reviewer. We will make our final decision based on the reviewer's comments.**

We look forward to receiving your revised manuscript.

Kind regards,

Xiaosheng Tan

Academic Editor

PLOS ONE

Reviewers' comments:

Reviewer's Responses to Questions

**Comments to the Author**

1. If the authors have adequately addressed your comments raised in a previous round of review and you feel that this manuscript is now acceptable for publication, you may indicate that here to bypass the “Comments to the Author” section, enter your conflict of interest statement in the “Confidential to Editor” section, and submit your "Accept" recommendation.

Reviewer #1: All comments have been addressed

Reviewer #2: All comments have been addressed

Reviewer #3: (No Response)

2. Is the manuscript technically sound, and do the data support the conclusions?

Reviewer #1: Yes

Reviewer #2: Yes

Reviewer #3: (No Response)

3. Has the statistical analysis been performed appropriately and rigorously? 

Reviewer #1: Yes

Reviewer #2: Yes

Reviewer #3: (No Response)

4. Have the authors made all data underlying the findings in their manuscript fully available?

Reviewer #1: Yes

Reviewer #2: Yes

Reviewer #3: (No Response)

5. Is the manuscript presented in an intelligible fashion and written in standard English?

Reviewer #1: Yes

Reviewer #2: Yes

Reviewer #3: (No Response)

6. Review Comments to the Author

**Reviewer #1: **The requested modifications have been scrupulously carried out.

These requested modifications were, in the majority of cases, necessary to allow greater clarity in the tools used to carry out this study. Corrections to the material and methods section have been carefully drafted to increase the clarity of the technique used in this manuscript.

The changes requested in the “Supporting info section” have been made, as have the minor modifications requested.

I recommend publication of this manuscript.

**Reviewer #2:** Thanks the authors for your thoughtful responses. All of my comments have been fully addressed. The updated Figure 1 clearly demonstrates how the measurements were done.

**Reviewer #3: **But the authors are probably joking about measuring a metabolic profile with their method!

Another mistake. Sodium azide is an inhibitor of ATP synthase. Are the authors aware of this?

However, the oxygen consumption profiles with the data obtained from their experiments are not presented. Figure 1A is a graphical construction!!!

This is enough to consider the results not tenable.

7. PLOS authors have the option to publish the peer review history of their article (what does this mean?). If published, this will include your full peer review and any attached files.

Reviewer #1: No

Reviewer #2: No

Reviewer #3: No

---

## [Author Response · Author response to Decision Letter 1]

20 Sep 2024

Dear Editors, 

Thank you for the opportunity to submit a revised draft of our manuscript “Comprehensive characterization of mitochondrial bioenergetics at different larval stages reveals novel insights about the developmental metabolism of Caenorhabditis elegans.” Please see below, in blue text, our responses to each of the reviewer comments. All page numbers refer to the clean version of the revised manuscript. 

Reviewer #1: 

The requested modifications have been scrupulously carried out.

These requested modifications were, in the majority of cases, necessary to allow greater clarity in the tools used to carry out this study. Corrections to the material and methods section have been carefully drafted to increase the clarity of the technique used in this manuscript.

The changes requested in the “Supporting info section” have been made, as have the minor modifications requested.

I recommend publication of this manuscript.

(No concerns to address)

Reviewer #2: 

Thanks the authors for your thoughtful responses. All of my comments have been fully addressed. The updated Figure 1 clearly demonstrates how the measurements were done.

(No concerns to address)

Reviewer #3: 

But the authors are probably joking about measuring a metabolic profile with their method!

We are not sure we understand. We do not mention a “metabolic profile.” The only time we use the word profile” in the manuscript is in this sentence: “To establish the profile of C. elegans mitochondrial bioenergetics throughout different life stages we stage-synchronized wild-type (N2) nematodes (Figure 2A, 2B) and measured OCR levels with and without different drugs (electron transport chain inhibitors and a mitochondrial uncoupler).” In this case, “profile” simply refers to “characterization.”

Another mistake. Sodium azide is an inhibitor of ATP synthase. Are the authors aware of this?

We are not clear what the reviewer describes as a mistake. We are aware that sodium azide is (also) an inhibitor of ATP Synthase, and so state in the manuscript (Lines 438-44): “Non-mitochondrial OCR is measured by inhibiting all oxidative phosphorylation using sodium azide, a potent complex IV/V inhibitor.”

However, the oxygen consumption profiles with the data obtained from their experiments are not presented. Figure 1A is a graphical construction!!! This is enough to consider the results not tenable. Yes, Figure 1A is a graphical illustration of how values were calculated and terms defined, as was requested by Reviewer 2. We consider (and note that Reviewers 1 and 2 appear to agree) that our Results are clearly shown, and Conclusions fully supported, by the Figures and data presented. As stated in our previous Response to Reviewers (9/10), “all the OCR curves (but not full profiles) are presented within the supplementary files.” Showing full OCR profiles for every experiment would require dozens of additional Figures, and would add almost no value beyond that provided by the raw data, all of which is provided in Supplemental (S1 File).

---

## [Editor Report · Decision Letter 2]

15 Oct 2024

Comprehensive characterization of mitochondrial bioenergetics at different larval stages reveals novel insights about the developmental metabolism of Caenorhabditis elegans

PONE-D-24-25658R2

Dear Dr. Meyer,

We’re pleased to inform you that your manuscript has been judged scientifically suitable for publication and will be formally accepted for publication once it meets all outstanding technical requirements.

Kind regards,

Xiaosheng Tan

Academic Editor

PLOS ONE
---

## [Editor Report · Acceptance letter]

22 Oct 2024

PONE-D-24-25658R2 

PLOS ONE

Dear Dr. Meyer, 

I'm pleased to inform you that your manuscript has been deemed suitable for publication in PLOS ONE. Congratulations! Your manuscript is now being handed over to our production team.

Kind regards, 

on behalf of

Dr. Xiaosheng Tan 

Academic Editor

PLOS ONE